# Source-Based EEG Neurofeedback for Sustained Motor Imagery of a Single Leg

**DOI:** 10.3390/s23125601

**Published:** 2023-06-15

**Authors:** Anna Zulauf-Czaja, Bethel Osuagwu, Aleksandra Vuckovic

**Affiliations:** Biomedical Engineering Research Division, School of Engineering, University of Glasgow, Glasgow G12 8QQ, UK

**Keywords:** EEG, sLORETA, neurofeedback, motor imagery, single leg

## Abstract

The aim of the study was to test the feasibility of visual-neurofeedback-guided motor imagery (MI) of the dominant leg, based on source analysis with real-time sLORETA derived from 44 EEG channels. Ten able-bodied participants took part in two sessions: session 1 sustained MI without feedback and session 2 sustained MI of a single leg with neurofeedback. MI was performed in 20 s on and 20 s off intervals to mimic functional magnetic resonance imaging. Neurofeedback in the form of a cortical slice presenting the motor cortex was provided from a frequency band with the strongest activity during real movements. The sLORETA processing delay was 250 ms. Session 1 resulted in bilateral/contralateral activity in the 8–15 Hz band dominantly over the prefrontal cortex while session 2 resulted in ipsi/bilateral activity over the primary motor cortex, covering similar areas as during motor execution. Different frequency bands and spatial distributions in sessions with and without neurofeedback may reflect different motor strategies, most notably a larger proprioception in session 1 and operant conditioning in session 2. Single-leg MI might be used in the early phases of rehabilitation of stroke patients. Simpler visual feedback and motor cueing rather than sustained MI might further increase the intensity of cortical activation.

## 1. Introduction

Recent advances in Brain Computer Interface (BCI) technology have led to the development of a range of neurorehabilitation applications that can be used in different environments, from clinics to patients’ homes. However, most of these systems are designed for motor rehabilitation of the upper limbs [1].

On the contrary, few studies have focused on the identification of movements of the lower limbs [2,3]. However, lower-limb neurorehabilitation is a major issue in clinical settings in people suffering from hemiplegia due to stroke [4]. Constraint-induced therapy is routinely used in rehabilitation of the upper limbs in people with stroke, to prevent the unaffected limb from overtaking cortical representation of the affected one [5]. For analogous reasons, it would be useful to isolate the motor cortex of the affected leg only for motor imagery practice early after stroke when overt movement may not be possible. Alternatively, a single-leg motor imagery (MI) BCI could be used as a priming tool prior to physical practice or could be combined with functional electrical stimulation or a robotic device in a closed-loop BCI, such as upper-limb applications in rehabilitation [6].

Designing a BCI system for neurorehabilitation of the lower limbs is a challenging task because not only are the cortical representations of the left and right leg closely located, but their corresponding motor cortex areas are located deep into the cortex [7] and many of the cortical structures are recruited by motor actions of either leg [8,9]. It is therefore not surprising that a recent review of BCIs based on the identification of lower-limb motor tasks [3] showed that most BCI systems identity just the first step, i.e., initiation of walking irrespective of the leg, which is typically used for control of the exoskeleton. Only very few studies have focused on the identification of movement of a single limb [3]. Several research groups attempted to classify left vs. right leg movement based on EEGs [10,11], achieving up to an 86% classification accuracy in an offline setting. Lee et al. [12] provided sensorimotor rhythm training over electrode location Cz for a group of stroke participants, showing improvements in dual task performance, walking speed, and balance. That was despite the participants being asked to imagine activities that did not specifically involve the legs, e.g., bowling, roller coaster, and boat racing. They provided limited information about the concomitant brain activity during the tasks. The only study that identified left vs. right movement in real time was based on fMRI. Mihara et al. [13] performed a randomized-controlled-trial fRMI neurofeedback of the lower limbs (instructions were to walk, without aiming, to detect the activity of a single leg) in people with stroke who demonstrated an improvement in gait function. However, fMRI has several disadvantages: it is expensive and bulky; it requires a person to relax, which is not optimal for motor neurorehabilitation; and it may be difficult to control due to the inherent delay [14].

A BCI based on an approximation of deeper cortical sources could be a relatively inexpensive proxy for fMRI. A 19-channel low-resolution sLORETA neurofeedback [15] has been used to treat adult Attention-Deficit Hyperactivity Disorder, cancer pain, or depression [16,17,18]. Other methods of source localization have also been implemented in custom-made neurofeedback systems. A neurofeedback system based on blind source separation based on up to 42 EEG electrodes in the OpenVibe platform has been used on healthy people for the improvement of spatial memory [19].

To be able to detect the motor cortex of one leg only, a larger spatial precision is required. Gu et al. [20] created an sLORETA source estimation based on a 64-channel EEG and built an offline classifier for left vs. right motor imagery based on phase locking values between different cortical regions. Abeln et al. [21] created an sLORETA model based on a 32-channel EEG to estimate different levels of forces exerted by isometric leg extension at different intensities but also performed the analysis offline. To the best of our knowledge, there has been no real-time detection of single leg movement based on source localization.

In our study, we present a custom-made sLORETA neurofeedback software application based on a 48-channel EEG and a customized graphical user interface developed specifically for leg motor imagery. The aim of the study is to test the feasibility of the neurofeedback system based on motor imagery of a single leg. The specific objectives are:To investigate the activation of brain structures due to single-leg MI with neurofeedback at the source level and how these compare to motor imagination and motor execution without feedback.To examine how lateralization in the sensory-motor cortex is affected by neurofeedback and compare it with the laterality during imagined and executed movement of a single leg without feedback.

## 2. Materials and Methods

### 2.1. Participants

Ten able-bodied participants completed the study (7 female, 3 male), the mean age was 28.4 ± 8.4 years, and all self-reported as right-handed and had no known mental or motor deficits. All participants signed an Informed Consent Form prior to taking part.

### 2.2. Experimental Setup

The EEG and EMG were recorded using the g.tec gUSBamp (Guger Technoloiges, Graz, Austria) bioamplifier with 44 passive EEG electrodes used from the locations of the international 10–20 system [22] covering the sensorimotor cortex and frontal, parietal, and occipital areas. The ground electrode was at AFz and a linked ear reference (A1 and A2) was used. The impedance was kept under 5 kΩ. The sampling frequency was set to 128 Hz and the signal was hardware-bandpass-filtered between 0.5 and 60 Hz and notch-filtered at 50 Hz.

Muscle activity was measured on the participant’s right leg using bipolar EMG over the tibialis anterior muscle and lateral head of the gastrocnemius with the reference and ground over the left and right ankle bones of the same leg. The sole purpose of EMG recording was to monitor muscle activity during motor imagery (MI) in order to exclude trials with a visible activity.

A Simulink model collected and saved data from the bioamplifier, under Matlab 2015a (Mathworks Inc., Natick, MA, USA). A custom neurofeedback software, rtLoretaNFB, enabled feedback from the selected source structure deeper in the brain. The rtLoretaNFB program had three parts: EEG data collection and parcellation, transformation of EEG channel data into the source space using the sLORETA software, and extracting values in defined ROIs. In addition, there were three ways of displaying the neurofeedback. These comprise a cross-section of the 3D model of the cortex (*slice view*), a view of the 3D cortical surface (*cortical surface*), and a bar plot (*bar*), as shown in Figure 1. A custom-made Graphical User Interface (GUI) was developed in C++ for the presentation.

The EEG was processed in a Simulink model that performed a running average over 128 samples (1 s) and saved each smoothed 128-sample segment to a text file. The directory where these one-second EEG segments were saved was monitored by rtLoreta NFB every 250 ms. One sLORETA brain volume was created per time point, and all 128 volumes were averaged to create a single volume for 1 s. For bar visual feedback, all voxels over a selected ROI were averaged. The single data processing loop took up to 250 ms, which meant that a total delay was approximately 1.25 s for neurofeedback.

The experimenter could choose between three (slice view, cortical surface, and bar plot) GUIs to show the activity of the selected ROI. Data from the sagittal view of the 3D cortex showing the left hemisphere from the interhemispheric fissure (including the motor cortex of the right leg) were used to provide visual feedback in this study.

### 2.3. Experimental Protocol

Participants’ motor imagery ability was assessed using the Kinaesthetic and Visual Imagery Questionnaire (KVIQ) [23]. Each participant was instructed to perform a specific movement, and then subsequently to imagine performing the movement. Movement involved the upper and lower limbs, head, and trunk. Visual MI performance was followed by kinaesthetic MI. During visual MI, participants were instructed to visually imagine a movement from the first-person perspective, and during kinaesthetic MI, they were asked to imagine sensations in the body during the corresponding movement, then to self-assess the quality of both imagery types on a 5-point Likert scale, 1 being the worst and 5 the best performance. The individual scores for the imagination of different parts of the body were averaged for an MI score for each participant, and across participants for an overall condition score.

Following this assessment, two EEG sessions were completed, at least one week apart. Session 1 comprised ME and MI without neurofeedback. Neurofeedback was provided during session 2 only. During both sessions, each participant was seated comfortably, approximately 1 m from the computer screen. The screen displayed a fixation cross and instructions, namely ‘Relax’ or ‘Imagine’. In session 2, the ‘Imagine’ instruction was followed by visual feedback, in a selected frequency band. The feedback was provided from the slice view and participants were explained which area of the slice corresponded to the primary motor cortex of the legs.

The frequency band was selected based on offline analysis of EEG activity during MI in session 1, for each individual, and corresponded to one of the pre-selected bands (8–15 or 16–24 Hz) with the strongest activity. Bands were determined based on our previous experience with MI tasks and roughly correspond to the alpha/low beta and beta sensory-motor rhythms [24].

The tasks during each session are shown in the order they were performed in Figure 2a,b, and are defined as follows:*Baseline* (BL)—Initial baseline data were acquired while a participant was sitting relaxed: two trials of 130 s with eyes open (EO) focusing on a fixation cross, and one of the same length with eyes closed (EC) in the following order—EO, EC, and EO. This task was the same in both Session 1 and 2, and it is shown in row 2 in Figure 2b.*ME*—The initial baseline was followed by the motor execution task where a subject was instructed to move the right leg by alternating the plantar flexion and dorsiflexion of the ankle when cued for movement: four trials of 130 s with alternating 20 s blocks of ME and resting, as shown in row 1 of Figure 2b. This task was the same in both Sessions 1 and 2.*MI1—*Following motor execution, the participant was asked to perform kinaesthetic MI by imagining the feeling of moving their right leg in the same manner as during the motor execution in the previous trials: four trials of 130 s with alternating 20 s blocks of MI and resting, as shown in row 1 of Figure 2b. This task was the same in both Sessions 1 and 2.*MI2* and *MI3—*In Session 1, this task did not differ from MI1. However, in Session 2, a participant performed the task as in MI1 but additionally received visual neurofeedback. It comprised ten trials of 130 s with alternating 20 s blocks of MI and resting.*MI4*—A participant was free to choose the type of right-leg kinaesthetic MI to perform without neurofeedback: four trials of 130 s with alternating 20 s blocks of MI and the resting state are shown in row 1 of Figure 2b. This task was the same in both Session 1 and 2.*Baseline* (final)—The final baseline task was similar to the initial baseline task. Three trials of 130 s were acquired. The task was the same in both Session 1 and 2, and it is shown in row 2 of Figure 2b.

### 2.4. Offline Data Analysis

#### 2.4.1. EEG Pre-Processing

The EEG for each subject was pre-processed in EEGLab [25]. Sections with high-amplitude noise larger than 100 μV were manually removed based on visual inspection. Independent Component Analysis (ICA) with the Infomax algorithm implemented in EEGLab was utilized. Common ICA weights and dipoles were calculated for a representative sample dataset containing resting, MI, and ME parts. The ICA components related to blinking, channel noise, or muscle artefacts were rejected. Once the EEG data were reconstructed from the remaining components, it was re-referenced to an average reference and any channels initially rejected were interpolated.

Depending on the number of noisy segments within the data for each participant, the maximum length of a clean EEG during each condition was between 150 s and 170 s. The maximum clean length of data for each condition was split into one-second epochs and the same number of one-second epochs of the initial baseline were used to compare baseline (BL) and action (MI or ME) intervals. The initial rather than between-trials baseline was used because, in prolonged repetitive movements, it might take a while for brain activity to return to the baseline value [26].

#### 2.4.2. Source Localization

Cleaned data were exported into the sLORETA program, which was used to derive the activity source location within the 3D brain space from the sensor-level EEG. The estimated current density sources, calculated using the linear minimum norm inverse solution, were mapped onto a 3D brain space with 6239 voxels with a 5 mm spatial resolution [27].

The MI and ME data as one-second epochs were tested using a t-test on log-transformed data, for normalization, to obtain significantly active voxels during each condition as compared to the baseline. The baseline consisted of a maximum of 180 s (or maximum usable length as described above) of the initial baseline for each participant, and the test condition was set to the same total length of data of MI (with and without neurofeedback) or ME.

The two frequency bands used were alpha/low-beta 8–15 Hz (called alpha further in the text) and mid-beta 16–24 Hz (called beta further in the text). Each of these produced one 3D image containing significantly active voxels during the condition in question (MI or ME) as compared to the initial baseline. From these, further analysis was performed on the chosen ROIs, which are related to movement and motor imagery.

The ROIs chosen for the offline analysis consisted of whole single-hemisphere BAs. The chosen BAs relating to somatosensory processing were the primary somatosensory cortex; BA 1–3; BA 5 pertaining to somatosensory processing; BA 40, which has been shown to be involved in proprioception and memory [26,28]; and BA 43, which is part of the secondary somatosensory cortex involved in sensory integration and proprioception [29].

Other movement-related areas included in the analysis are the primary motor cortex, BA 4, and the premotor cortex and supplementary motor area (SMA) in BA 6. Other movement-related areas also included are BA 7, responsible for visual-motor coordination during movement; BAs 9 and 10 related to motor planning, memory, and decision making; and BA 24 that is a part of the ventral anterior cingulate cortex, which has been associated with action–outcome learning [30] in addition to other functions such as cognitive functions, pain, and attention roles.

Finally, we included BA 17 that contains the primary visual cortex for processing of visual stimuli in order to inspect the effects of observing a target during neurofeedback.

#### 2.4.3. Lateralization Index and Percentage of Active Volumes

For each frequency band, active voxels were summed up within each of three areas BA 4, BA 6, and BA 1–3, over each hemisphere separately, and a weighted laterality index (wLI) value was calculated according to Equation (1) for each of these three areas [31]. These specific areas were chosen because they exhibit cortical mapping of body parts on the contralateral side; hence, the laterality of activity could potentially be used to distinguish between left and right lower-limb activity.
(1)$$wLI=\frac{\sum v_{L}^{w}-\sum v_{R}^{w}}{\sum v_{L}^{w}+\sum v_{R}^{w}}$$
where $\sum v_{L}^{w}$ and $\sum v_{R}^{w}$ are the sum of all active voxels in the chosen ROI of the left and right hemispheres, respectively. Each voxel is weighted according to its statistical significance value so that the voxels that have strong activity contribute more than those with weak activity [32]. Left (contralateral) dominance is defined as wLI > 0.2, right (ipsilateral) dominance as wLI < −0.2, and bilaterality as |wLI| < 0.2 [32].

Additionally, the percentage of ROI active during each condition was calculated according to Equation (2).
(2)$$p=\frac{v_{a}}{v_{ROI}}\times 100\%$$
where *v_a_* is the number of active voxels in the given ROI and *v*_ROI_ is the total number of voxels in the whole ROI. This calculation was performed for BAs relating to the primary and supplementary motor and sensory cortex, that is, BA 1–3, 4, 5, 6, 40, and 43.

The changes in wLI and p were compared between conditions and sessions.

#### 2.4.4. Statistical Analysis

In order to obtain statistically significant active voxels in sLORETA, the paired t-test was performed on log-transformed data to compare each condition to the initial baseline for the group of 10 participants. The significance level was set to *p* < 0.05.

## 3. Results

Motor imagery scores (KVIQ) are presented, followed by the results of the analysis of differences between motor actions (ME and MI with and without feedback) and baseline data, across different ROIs; this is followed by the analysis of wLI in selected ROIs.

### 3.1. Kinesthetic and Visual Motor Imagery

The mean kinaesthetic imagery (KI) score across participants was 3.27 (min 2.17, max 4.35), while for visual imagery (VI), it was 3.72 (min 2.17, max 5.00). This shows that participants had on average medium motor imagery ability, which was similar for the visual and kinesthetic imagination.

### 3.2. sLORETA Volume Analysis

In order to compare the activity through source localization, 3D brain images were created for each condition. Figure 3 shows active areas for (i) ME and for (ii) joint MI2 and MI3 for both sessions (S1 on day one and S2 on day two). Recall that during S1, motor imagery MI2 and MI3 were without feedback, and during S2, feedback was provided. We show results for two frequency bands, corresponding to frequency bands that were used for neurofeedback. For each participant, neurofeedback was based on the frequency band that showed the strongest event-related desynchronization during motor execution. For eight participants, it was 8–15 Hz, and for two, it was 16–24 Hz. Figure 3 shows active areas from the medial view of the left and right hemispheres from the interhemispheric fissure. Areas more active during ME/MI compared to the baseline are graded from red to yellow, and areas in blue show where the baseline is more active.

Generally, it can be seen that the cortical activities were similar in patterns in both hemispheres. During ME, there was more activity in the primary sensorimotor cortex in both frequency bands and both sessions with respect to the baseline, while the occipital and frontal areas were more active during the baseline than ME.

During MI, the occipital area was still more active for the baseline in Session 1 where no feedback was used, as demonstrated in the left hemisphere. This was not the case in Session 2 where the occipital activity increased, best represented in Figure 3F. The change in the occipital activities for Session 2 is likely related to the processing of the visual feedback, which was used in this session.

The frontal area activation during MI varied, with higher activity during session 1 in the 8–15 Hz band, but higher baseline activity for session 2 in the same area in the left hemisphere. This pattern of activity is somewhat reversed in the right hemisphere.

Finally, the premotor cortex was moderately active during MI in session 1 in the 8–15 Hz band, and in session 2 in the 16–24 Hz band only. This region is similar to the cortical area active during ME. It can be observed that MI with and without feedback activates different cortical areas; MI without feedback activates dominantly the prefrontal cortex relevant to motor planning, while MI with feedback activates the primary cortex similar to ME. This indicates possibly different mental strategies applied for MI in sessions S1 and S2.

From the source projections, the voxels that were significantly more active during MI2 and MI3 than during the initial baseline of each session were found and assessed for several ROIs in the primary and supplementary motor and sensory cortices (Table 1). Results are shown for both S1 and S2, for left and right hemispheres separately.

These results show that primary and secondary motor and sensory areas were less active in the 8–15 Hz band when neurofeedback was provided; however, a moderate increase in activity in these areas was observed in the 16–24 Hz band with a shift to the ipsilateral hemisphere.

Table 1 shows that in the 8–15 Hz band during MI without neurofeedback, sensorimotor areas had significant active volumes with relative contralateral dominance with regard to the number of active voxels. During MI with neurofeedback in S2, these same areas presented a marginal degree of activity in the 8–15 Hz band in most BAs, and none in BA 9, 10, 24, and 43. Notably, BA 7 exhibited an increase in the active volume with respect to MI without feedback in S1. Moreover, BA 17 showed significant activity, which may be attributed to the visual processing of neurofeedback received,

In the 16–24 Hz band, most of the sensorimotor areas were active during MI alone in S1 but at smaller volumes than in the alpha band, and hemispheric dominance varied between the BAs, with the primary somatosensory and motor cortex (BA 1–3, 4) and premotor cortex (BA 6) exhibiting right dominance (ipsilateral). With the introduction of neurofeedback in S2, these ipsilateral areas generally increased in activity, judging by the number of active voxels. In addition, BA 5, 7, and 40 presented higher activity in S2 also in the ipsilateral hemisphere. No activity was observed in BAs 9, 10, 17, and 43 during neurofeedback in this frequency band.

Based on these results, BA 4 representing the primary sensorimotor cortex, BA 6 as the pre-motor, and SMA were selected for more detailed analysis as these areas contain cortical mapping of body parts. In addition, BA 40 and 43 were also included as these areas generally presented a high degree of activity during MI trials in both frequency bands and sessions. The following radar plots show results of this volume analysis, with the percentage of volume of the ROIs active during each of the conditions for Session 1 and 2 during ME and MI1-4. To remind the reader, sub-sessions MI1 and MI4 were performed without neurofeedback in both S1 and S2. Figure 4a shows active volumes in the 8–15 Hz band, while Figure 4b shows active volumes in the 16–24 Hz band.

Figure 4a shows that the active volumes during ME for sessions S1 and S2 were comparable, confirming the intersession repeatability and ME as a suitable reference for brain activities during MI. Active volumes within BA 4, 6, 1–3, and 40 varied between approximately 20 and 60% of each ROI throughout session 1 trials (blue). The highest values were seen in ME and MI4 conditions. In each ROI, the active volume generally decreased in MI1 as compared to ME, and then rose with each subsequent trial to MI4, indicating learning. The active volumes for BA 43, which is related to proprioception with activities that could indicate the quality of MI, were much higher than other ROIs for MI1, MI2, and MI4 in session 1 as the percentage of active voxels was up to 100%.

During ME in session 2 (Figure 4a in orange), a slightly higher degree of activity can be observed, with peaks in BA 4, 6, and 1–3 exceeding the corresponding points for session 1; however, the active volumes for MI1-MI4 were generally much lower than those during session 1 with several conditions presenting an activity below 10% in a particular ROI. For instance, this was the case for BA 43 in session 2, which showed 19% activity in MI4 only and zero in other conditions (Figure 4a). The low or lack of activation during MI2 and MI3 in session 2 (Figure 4a) may be explained by the neurofeedback provided during these trials, causing other brain structures (e.g., visual cortex) to be more involved compared to those presented above. Interestingly, however, this low activation was also observed during MI1 in S2, which did not receive neurofeedback; therefore, it suggests that the level of motor and sensory activity during S2 may have generally been lower in the 8–15 Hz band than during S1 possibly due to the familiarity of a simple task, which was no more engaging in S2.

Figure 4b shows the activity in 16–24 Hz. For session S1 (blue), there was no consistent increase in activation from M1 to M4. However, for S2 (orange), in all BA except BA43, there was a consistent increase in activation from M1 to M4, which potentially indicates that the participant may have been improving on the task performance. MI subsections in orange in Figure 4b generally present a higher active volume than in Figure 4a. This suggests that neurofeedback may have increased the activity in BA 4, 6, and 1–3 in the 16–24 Hz band, but not in 8–15 Hz, and is consistent with Table 1.

In summary, the learning effect over MI1-4 in sessions without neurofeedback was present in the alpha band, while in sessions with neurofeedback, it was observed in the beta band.

### 3.3. Laterality Index during Real and Imagined Movements

The laterality index wLI is presented for BA 4, 6, and 1–3 for both 8–15 Hz (Figure 5a) and for 16–24 Hz (Figure 5b). Figure 5a shows that the weighted laterality index was highly dependent on the active volume within each ROI. Just as session 1 in Figure 4a showed a consistent active volume for each ROI, the corresponding wLI also showed consistency in session 1 (Figure 5a blue). In 8–15 Hz, a consistent, bilateral to moderately contralateral wLI (with the value varying between approximately 0 and 0.4) was present through all sub-sessions of Session 1.

During session 2, those trials with a similar level of active volumes as in session 1 showed a similar range of wLI values, for instance, ME and MI4 in BA 4 with wLI = 0.13 and 0.07, respectively; ME and MI4 in BA 6 both with wLI = 0.37; and ME in BA 1–3 with wLI = 0.14 (Figure 5a). In contrast, trials with a low active volume during session 2 showed extreme LI values, which may not be reliable.

Figure 5b shows the wLI for BA 4, 6, and 1–3 in the 16–24 Hz band. Since the active volumes in sessions 1 and 2 in this band were more comparable than in the 16–24 Hz band, the wLIs also showed more similarity, with session 1 values ranging between −0.8 and 0.6, and session 2 values ranging between −1 and 0.5. Again, in this case, the closer the values of volume of the ROI are for both sessions, the closer the wLI values are; for instance, active volumes in BA 6 for MI4 were 21% for session 1 and 38% for session 2, while the respective wLI values were −0.03 and 0.13 for session 1 and 2; hence, both exhibited bilaterality in this case.

## 4. Discussion

The results of the study show that the main effect of real-time neurofeedback in the form of a 3D brain model was a small increase in the activity of the primary sensorimotor cortex in the beta band, while MI without neurofeedback resulted in activation of the supplementary motor cortex.

Considering the first objective, to compare active ROIs during MI of a single leg with and without neurofeedback, in the 8–15 Hz band during neurofeedback, there was less activity in the sensory motor structures as compared to MI without feedback. This was a surprising result, especially since most participants received neurofeedback in this frequency band. This may be partially explained by the effect of neurofeedback in the 8–15 Hz band being overshadowed by visual components and fatigue, which predominantly take place in this band [33,34]. During MI with neurofeedback, the activity increased in the beta band.

Activation of SMR in different frequency bands for different MI modalities may be explained by different motor strategies of MI with and without neurofeedback. Most notably, during MI without feedback, participants were likely more focused on the kinesthetic (proprioceptive) aspect, and during MI with feedback, they had to share the attention between the screen and MI. Alpha and beta sensory-motor rhythms have different sources [35]. Ritter et al. found that alpha ERD is located posteriorly with respect to beta ERD, bringing them to the conclusion that Rolandic alpha rhythm is linked predominantly to sensory processing while Rolandic beta rhythm is linked predominantly to motor processing [35]. In addition, in an MEG study, Jurkiewicz et al. [36] also confirmed that beta rebound originates in the motor cortex. Thus, a stronger sense of proprioception in MI without feedback might have resulted in stronger 8–15 Hz band activity, while feedback might have reduced the focus on proprioception. As a result, the motor component visible as 16–24 Hz activity became dominant. In support of this, the spatial distribution of activity in the ROI in the 16–24 Hz band was similar to that during ME, covering the primary motor cortex.

A simpler neurofeedback mode, such as a bar, used in our pilot study [37] would possibly have required less visual processing than the 3D cross-section and the effect of neurofeedback could have been clearer. Nevertheless, there was some effect of learning during the neurofeedback session exhibited by the increased level of activity in many of the BAs during MI4, which no longer had access to neurofeedback.

A noteworthy result is that the prefrontal cortex was activated during MI alone, which is consistent with other MI studies [38,39]. On the other hand, MI with neurofeedback activated the central regions corresponding to the motor cortex that participants were instructed to modulate. This suggests that people performed MI when neurofeedback was not present and switched to the operant conditioning strategy (increasing activity of brain areas shown on the screen, in this case, primary motor cortex) when neurofeedback was provided.

The neurofeedback effect might be stronger if shorter sequences of cue-based MI were reinforced. Previous studies showed that during prolonged sustained MI, ERD disappears within 4–5 s and, for that reason, ERD during short, cued MI is stronger than during self-paced sustained MI [40]. Due to a small number of 20 s sub-sessions, we could not analyze only the first 4–5 s of MI to compare the cortical activity with the one during cue-based MI.

The reason that we asked participants to perform a sustained MI for 20 s was to mimic as closely as possible a typical motor task in fMRI BCI studies, where sustained MI can last up to 60 s [14]. The 20 s duration was a compromise of mimicking fMRI BCI and avoiding participants becoming tired and losing concentration.

Another factor that might have caused an increase in the beta activity during alpha neurofeedback is the existence of a general neurofeedback network, which is believed to be active irrespective of the neurofeedback protocol [41]. The existence of this network was confirmed based on fMRI studies that could not analyze the frequency content, so it is possible that some structures of this network are active in the alpha while others are active in the beta band. Based on a meta-analysis, Emmert et al. [41] suggested that the activity in areas such as the anterior insular cortex, basal ganglia, occipital cortex, prefrontal cortex, and parieto-temporal junction increases, while the activity in areas such as the anterior cingulate cortex decreases during neurofeedback. In this study, the activity of the anterior cingulate cortex (BA24) was notably decreased in the 8–15 Hz band during neurofeedback. Likewise, the activity of the occipital cortex increased in the same band while the activity of parietal and tempo-parietal regions (BA 5, 7,40) increased in the 16–24 Hz band.

With regard to the second objective, the laterality index was found to be affected by neurofeedback but this effect was unstable. Because wLI was highly dependent on having a sufficient active volume for analysis, the resulting values were possibly unreliable. In the 8–15 Hz band, wLI was consistent across sub-sessions of session 1, none of which included neurofeedback. These results showed predominantly contralateral or bilateral wLI, which is consistent with the known patterns of activation in the literature [42,43]. This was not the case for session 2; as previously mentioned, these structures no longer had much significant activity and, therefore, wLI could not be accurately estimated, leading to extreme values. In the 16–24 Hz band, the active volumes were generally above 10% of ROI for neurofeedback trials of session 2, and the corresponding wLIs were between -1 and −0.8, leading to right hemispheric dominance, which is consistent with ROI volume analysis. The fact that ME and MI without feedback show contralateral activity indicate that the sLORETA was able to precisely identify active structures.

A limitation of the study is that we calculated LI with respect to the whole left and right ROI, rather than areas corresponding to legs, but there are limited data from the literature that would allow a precise leg localization in sLORETA, and there were a limited number of active volumes in the current study. To compensate for this, we provided feedback from a cross-sectional area of the brain that included the primary motor cortex of the selected leg and instructed participants to focus on that area during neurofeedback. However, this feedback was too abstract and could have inadvertently distracted participants who would then forget to focus on MI as the main mental strategy. In our pilot study where feedback was provided in the form of a bar, contralateral activity was visible [37].

Most brain-computer interface systems based on MI are cue-based [44], and the purpose of the cue is to facilitate initiation of the movement, which also helps participants to maintain focus. In this study, the external reminder to perform MI was absent and it is likely that participants switched to operant conditioning, applying whatever strategy in order to control the visual feedback. In the future, feedback should be simpler to avoid distraction and should constantly remind the user to apply MI as a neurofeedback strategy.

Another limitation of this study is the short NF training time, due to lengthy experimental procedures, which involved MI with and without NF on the same day. It is known that mastering neurofeedback requires some practice, while in this study, there was only one session. It is possible that control would improve with practice, as we have noticed in sensor-level neurofeedback studies [45,46]. On the other hand, because it was necessary to compare both modalities of MI on the same day, the extended duration of a session would cause fatigue and would likely increase participant dropout rates.

There are many software packages that allow real-time source analysis such as MNE Scan [47], NFBLab [48], and Neuropype (Intheon, CA, USA). Some of these are open-source and some are commercially available. Recently, RT-NET [49] debuted in the field, claiming to provide an excellent temporal resolution with only 4 ms of delay. We decided to create our own system in order to have larger flexibility in creating a graphical user interface. A delay of 1.25 s is much smaller than that of a typical fMRI. A processing delay of 250 ms is comparable with results from the literature and a shorter overall delay could be achieved by reducing the 1 s time window from which features are currently extracted.

This study was performed with able-bodied participants who could successfully perform MI on their own as per the KVIQ results. In the case of patients with motor and sensory deficits, that ability to imagine movement of the affected limb might be compromised [50] and the benefit of visual feedback might be larger than in able-bodied people. In both EEG [11] and fMRI [12] neurofeedback studies with stroke patients, participants were asked to perform imaginary motor action bilaterally and, although motor improvement was achieved following prolonged neurofeedback, laterality of the motor imagination was not specifically investigated. Thus, it was not possible to compare these studies with our findings.

## 5. Conclusions

The present study demonstrated the effect of neurofeedback from source structure activity derived from scalp EEGs. The patterns of activation are generally consistent with the literature and between experimental sessions. MI with and without neurofeedback activates distinctively different cortical structures, indicating different mental strategies. The motor and sensory structures did not respond in the alpha/low-beta frequency band of feedback but an increase in activity was found in the mid-beta band, although the ipsilateral side and visual cortex were activated during neurofeedback trials. The laterality index was more stable in the session without feedback, showing bilateral activity. More neurofeedback sessions might be required to assess the effectiveness of neurofeedback. People who lack motor imagery activity due to neurological issues might benefit more from the visual feedback than the able-bodied people.

## Figures and Tables

**Figure 1 sensors-23-05601-f001:**
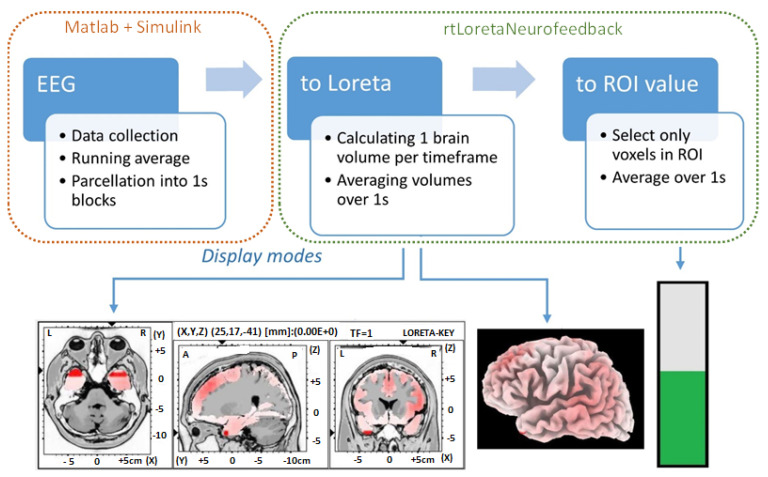
Flowchart representation of the custom neurofeedback software showing each of the three steps, EEG collection and parcellation, conversion to sLORETA brain volumes, calculating average value for user-specified ROI, and neurofeedback display modes: slice view, cortical surface, and bar.

**Figure 2 sensors-23-05601-f002:**
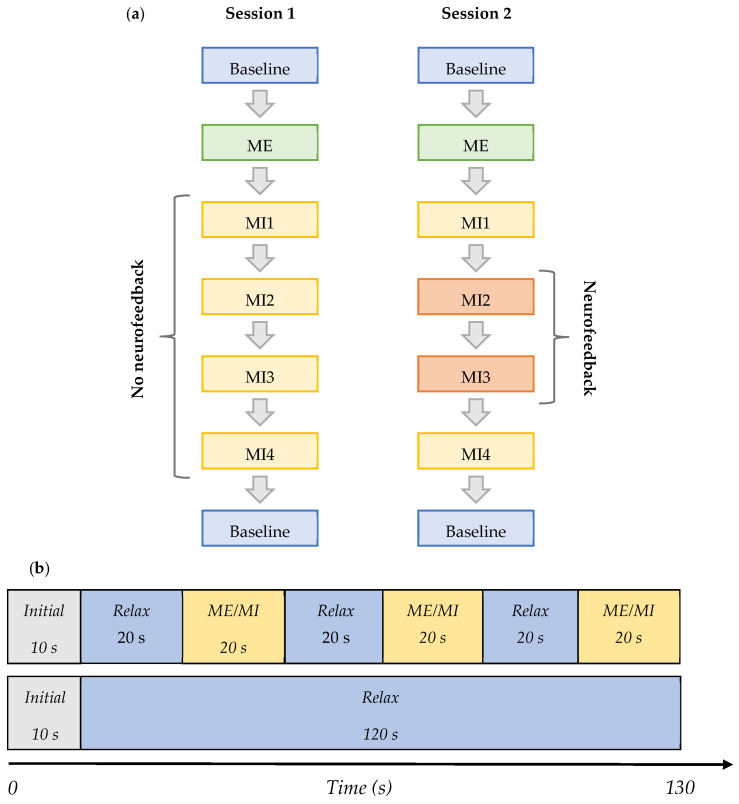
(**a**) Flowchart of the subsections of sessions 1 and 2. Each ‘Baseline’ block consisted of 2 baselines with eyes open and one with eyes closed. Each ME or MI block consisted of four trials of ME or MI, respectively, as shown in Figure 2. No neurofeedback was received in MI 1-4 during session 1 and in MI 1 and MI4 during session 2. Neurofeedback was provided during MI2 and MI3 of session 2 only. (**b**) Diagram showing the content of the ME/MI trials and baseline trials—both 130 s total.

**Figure 3 sensors-23-05601-f003:**
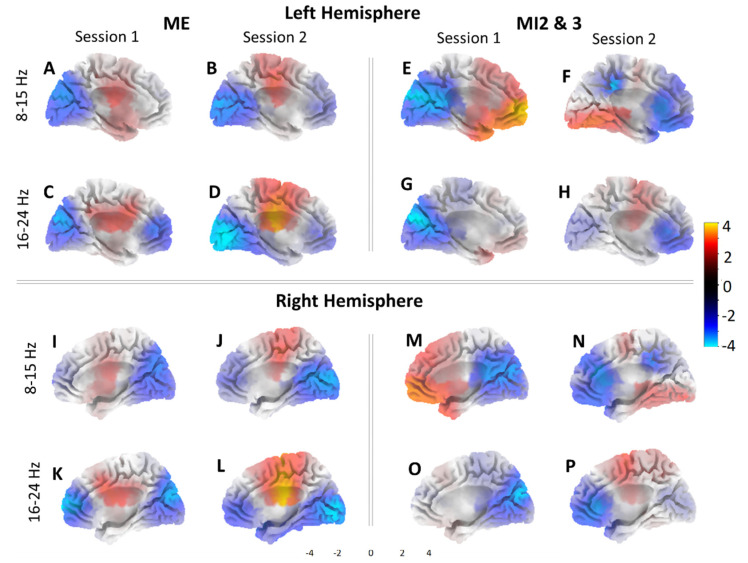
Source projections using sLORETA for the group of 10 participants. Left hemisphere during: ME in 8-15 Hz for S1 (**A**) and S2 (**B**), 16–24 Hz band for S1 (**C**) and S2 (**D**); MI2 and 3 in 8–15 Hz band for S1 (**E**) and S2 (**F**), 16–24 Hz band for S1 (**G**) and S2 (**H**). Right hemisphere shown during: ME in 8–15 Hz for S1 (**I**) and S2 (**J**), 16–24 Hz band for S1 (**K**) and S2 (**L**); MI2 and 3 in 8–15 Hz band for S1 (**M**) and S2 (**N**), 16–24 Hz band for S1 (**O**) and S2 (**P**). Red and yellow colors indicate greater activity during ME/MI compared to the baseline. Blue indicates greater activity for baseline. Neurofeedback was received during (**F**,**H**,**N**,**P**).

**Figure 4 sensors-23-05601-f004:**
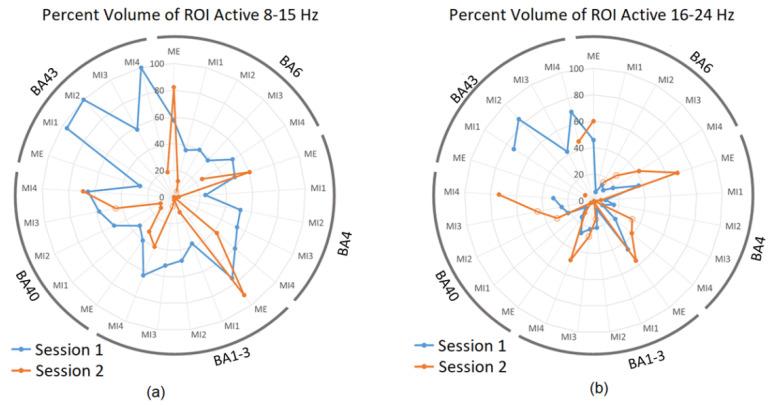
The percentage of each ROI active during session 1 and 2 in (**a**) 8–15 Hz band and (**b**) 16–24 Hz for selected BAs. Results are presented for each sub-session separately. Blue datapoints indicate volumes significantly more active during ME/MI as compared to baseline during session 1, and orange datapoints indicate those significantly more active during ME/MI than baseline in session 2. Open circle datapoints for session 2 indicate that neurofeedback was received during this trial. Axis 0 to 100%.

**Figure 5 sensors-23-05601-f005:**
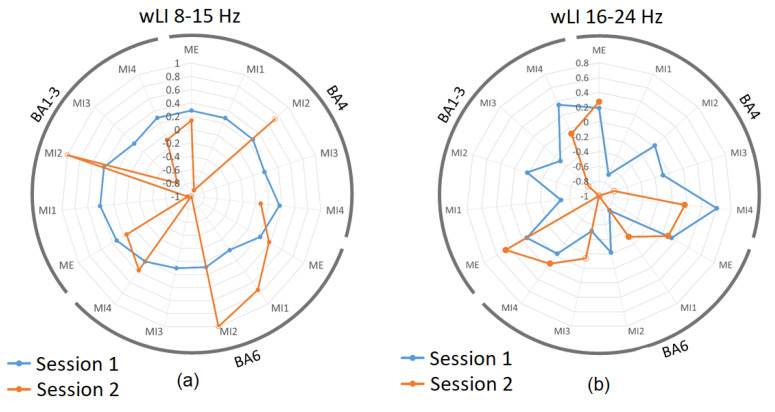
The Laterality Index of primary sensory (BA1–3), primary motor (BA4), and supplementary and premotor cortices (BA6) in sessions 1 and 2 in (**a**) 8–15 Hz band and (**b**) 16–24 Hz. Results are presented for each sub-session separately. Blue datapoints indicate volumes significantly more active during ME/MI as compared to baseline during session 1, and orange datapoints are those significantly more active during ME/MI than baseline in session 2. Open-circle datapoints for session 2 indicate that neurofeedback was received during this trial. Positive values correspond to contralateral while negative values correspond to ipsilateral activity; values between −0.2 and 0.2 correspond to bilateral activity.

**Table 1 sensors-23-05601-t001:** Percentage volume of each ROI significantly more active during sub-sessions MI2 and 3 vs. baseline in two frequency bands during session S1 (no feedback) and session 2 (feedback denoted with asterisk).

BA	Condition	% Volume Active 8–15 Hz				% Volume Active 16–24 Hz			
		S1		S2 *		S1		S2 *	
		L	R	L	R	L	R	L	R
1–3	MI2	30	18	2	0	10	11	0	14
	MI3	28	24	1	6	8	14	4	24
4	MI2	24	17	4	1	6	8	0	16
	MI3	22	16	0	0	5	6	3	23
5	MI2	9	0	7	0	7	0	0	15
	MI3	10	2	5	6	9	0	10	25
6	MI2	28	24	2	0	6	11	0	0
	MI3	27	27	0	0	1	6	15	18
7	MI2	2	0	10	0	2	0	0	12
	MI3	5	0	11	4	2	0	4	19
9	MI2	47	51	0	0	15	11	0	0
	MI3	47	45	0	0	0	5	0	0
10	MI2	50	49	0	0	0	0	0	0
	MI3	50	50	0	0	1	0	0	0
17	MI2	0	0	46	6	0	0	0	0
	MI3	0	0	31	0	0	0	0	0
24	MI2	25	18	0	0	0	0	0	0
	MI3	18	11	0	0	0	0	6	14
40	MI2	41	9	11	0	20	2	0	31
	MI3	40	18	11	34	17	8	7	37
43	MI2	54	46	0	0	50	35	0	0
	MI3	54	4	0	0	42	0	0	0

## Data Availability

Raw EEG data are available from the authors on reasonable request.

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
