# Peer review of "Source-Based EEG Neurofeedback for Sustained Motor Imagery of a Single Leg"

_sensors, 2023, doi:10.3390/s23125601_

Round 1

Reviewer 1 Report

Authors proposed to test the visual neurofeedback guided MI of dominant leg based on source analysis with real-time sLORETA. Discussion and analysis looks fine. However, some Figure quality need to be improved. Thus, there are some comments to be revised.

1. Figure 1 looks unclear to be seen. Please replace it with higher quality.

2. Please change the hyphen (-) into a minus sign (−) in Line 252.

3. Please change p<0.05 to p <0.05 due to broken grammar.

4. Please use clearer fonts in Figure 3 because some letters looks unclear to be seen.

5. Figures 4 and 5 labels are too small to be seen. Please enlarge the fonts.

6. Please correct Most Brain Computer Interface to Most brain computer interface.

7. Please provide ref. (Most Brain Computer Interface systems based~) with ref. (https://www.mdpi.com/1424-8220/22/16/6042).

Some English expression need to be corrected even though most of English expression looks fine. 

Author Response

Responses to Reviewer 1

Authors proposed to test the visual neurofeedback guided MI of dominant leg based on source analysis with real-time sLORETA. Discussion and analysis looks fine.

We would like to thank the reviewer on positive comments.

However, some Figure quality need to be improved. Thus, there are some comments to be revised.

  1. Figure 1 looks unclear to be seen. Please replace it with higher quality.

Answer: Figure replaced

  1. Please change the hyphen (-) into a minus sign (−) in Line 252.

Answer: replaces

  1. Please change p<0.05 to p <0.05 due to broken grammar.

Answer: a space added

  1. Please use clearer fonts in Figure 3 because some letters looks unclear to be seen.

Answer: Figure replaced

  1. Figures 4 and 5 labels are too small to be seen. Please enlarge the fonts.

Answer: Fonts enlarged

  1. Please correct Most Brain Computer Interface to Most brain computer interface.

Answer: Corrected

  1. Please provide ref. (Most Brain Computer Interface systems based~) with ref. (https://www.mdpi.com/1424-8220/22/16/6042).

Answer: Reference added

Comments on the Quality of English Language

Some English expression need to be corrected even though most of English expression looks fine. 

Answer: English revised throughout the text

Reviewer 2 Report

In this work, the authors tested the feasibility of visual neurofeedback guided motor imagery (MI) of the dominant leg. This manuscript designed two sessions to demonstrate the effect of neurofeedback from source structure activity derived from scalp EEG. Ten participants completed the experiments. Results showed that the neurofeedback might further increase the intensity of cortical activation. Some related studies could be discussed: Internal Feature Selection Method of CSP Based on L1-Norm and Dempster-Shafer Theory. IEEE Transactions on Neural Networks and Learning Systems. 2021,32(11): 4814-4825, Correlation-based channel selection and regularized feature optimization for MI-based BCI,Neural Networks,2019,118:262-270,  A brain–computer interface based on miniature-event-related potentials induced by very small lateral visual stimuli[J]. IEEE Transactions on Biomedical Engineering, 2018, 65(5): 1166-1175.However, there were some questions as follows:

1.     Why use a filtering range of 0.5-60Hz? Isn't the commonly used high-frequency cutoff in MI 30Hz or 40Hz?

2.     What was the reason for using 20s to perform the MI tasks? More comparisons on the time ranges should be given.

3.     The figures were too blurry to see. Please check these carefully.

4.     Sections with high amplitude noise larger than 100 uV were manually removed based on visual inspection. What about the time range?

5.     Areas more active during ME/MI compared to the baseline were graded from red to yellow, and areas in blue showed where the baseline was more active. What did the yellow and red colours represent?

6.     Whether the weighted laterality index (wLI) value was widely accepted? Please give more citations for this.

7.     What was the reason for using the threshold like ‘wLI>0.2’? Additionally, there were several user-defined parameters mentioned in this manuscript. How to determine these parameters? Please give more experiments to analyze these parameters.

Author Response

Response to Reviewer Comments

In this work, the authors tested the feasibility of visual neurofeedback guided motor imagery (MI) of the dominant leg. This manuscript designed two sessions to demonstrate the effect of neurofeedback from source structure activity derived from scalp EEG. Ten participants completed the experiments. Results showed that the neurofeedback might further increase the intensity of cortical activation. Some related studies could be discussed: Internal Feature Selection Method of CSP Based on L1-Norm and Dempster-Shafer Theory. IEEE Transactions on Neural Networks and Learning Systems. 2021,32(11): 4814-4825, Correlation-based channel selection and regularized feature optimization for MI-based BCI,Neural Networks,2019,118:262-270,  A brain–computer interface based on miniature-event-related potentials induced by very small lateral visual stimuli[J]. IEEE Transactions on Biomedical Engineering, 2018, 65(5): 1166-1175

Answer: We would like to thank the reviewer for the suggested references but we respectfully disagree that these references are relevant to the topic of this study. The first two references discuss novel classification algorithms and feature optimization algorithms while our study uses no classifier and no classification features. The third study is based on event-related potential while our study is based on event-related potentials while our study is based on source localisation of oscillatory brain activity/

.However, there were some questions as follows:

  1. Why use a filtering range of 0.5-60Hz? Isn't the commonly used high-frequency cutoff in MI 30Hz or 40Hz?

Answer: This was a hardware filter setting in the EEG device, there was no 40 Hz option, and 30 Hz would be too low because every filter inevitably affects neighboring (lower and higher) frequencies. We explained on line 100 that it was a hardware filter.

  1. What was the reason for using 20s to perform the MI tasks? More comparisons on the time ranges should be given.

Answer: We wanted to mimic experimental conditions for fMRI BCI because we simulated real time fMRI with real time sLORETA. On lines 463-470 we explained

Previous studies showed that during prolonged sustained MI, ERD disappears within 4-5s and for that reason ERD during short, cued MI is stronger than during self-paced sustained MI [27]. Due to a small number of 20s sub sessions, we could not analyse only the first 4-5s of MI to compare the  cortical activity with the one during cue based MI.

The reason that 20s we asked participants to perform a sustained MI was to mimic as closely as possible typical motor task in fMRI BCI studies, where sustained MI can last up to 60s [16]. The 20s duration was a compromise of mimicking fMRI BCI and avoiding participants getting tired and losing concentration. “

Please note that we were dividing each of the three 20s segments into 1s segments for sLORETA localisation and averaged them all, as explained is section 2.4.2. From that reason, is why if we analysed only the initial 4-5s we would have 4-5 less data to analyse.

  1. The figures were too blurry to see. Please check these carefully.

Answer: We replaced Figures 1,3,4 and 5

  1. Sections with high amplitude noise larger than 100 uV were manually removed based on visual inspection. What about the time range?

Answer: We assume that with the time range, the reviewer refers to the total duration of EEG recording after removing noisy segments. We explained this in section 2.4.1. that after removing noise, between 150s and 170s EEG was left for analysis for each participant.

  1. Areas more active during ME/MI compared to the baseline were graded from red to yellow, and areas in blue showed where the baseline was more active. What did the yellow and red colours represent?

Answer: we added a colour bar

  1. Whether the weighted laterality index (wLI) value was widely accepted? Please give more citations for this.

Answer: We provided a reference t Seghier, M.L.Laterality index in functional MRI: methodological issues.Magn Reson Imaging. 2008, 26(5):594-601. which is a review paper, showing that a number of studies used this equation

  1. What was the reason for using the threshold like ‘wLI>0.2’? Additionally, there were several user-defined parameters mentioned in this manuscript. How to determine these parameters? Please give more experiments to analyze these parameters.

Answer: the same reference Seghier et al, discuss the threshold values in different studies, showing that most studies adopt the value of ‘wLI>0.2’

The reviewer might also be concerned about source localisation, line 215  “3D brain space with 6239 voxels with 5mm spatial resolution [29]” These numbers are default sLORETA values provided in reference 29